Gene expression during THP-1 differentiation is influenced by vitamin D3 and not vibrational mechanostimulation

Simakou Theodoros 1 2
Freeburn Robin 1
Henriquez Fiona L. fiona.henriquez@uws.ac.uk 1
1 School of Health and Life Sciences, University of West of Scotland , Paisley , United Kingdom
2 Institute of Infection, Immunity and Inflammation, University of Glasgow , Glasgow , United Kingdom
Leppik Liudmila
Electronic publication date: 2021 Jul 14
Publication date: 2021
Volume: 9
Electronic Location ID: e11773
Received 2021 Feb 17; Accepted 2021 Jun 23
Copyright: ©2021 Simakou et al.
Copyright year: 2021
Copyright holder: Simakou et al.
License: This is an open access article distributed under the terms of the Creative Commons Attribution License, which permits unrestricted use, distribution, reproduction and adaptation in any medium and for any purpose provided that it is properly attributed. For attribution, the original author(s), title, publication source (PeerJ) and either DOI or URL of the article must be cited.
License URL: https://creativecommons.org/licenses/by/4.0/

Keywords: THP-1, Vitamin D3, Macrophage differentiation, PIEZO1, PKD2, Mechanosensation, TCF/LEF, Vibrational Stimulation

Funding: Alopecia UK and the University of West of Scotland This work was supported by Alopecia UK and the University of West of Scotland. The funders had no role in study design, data collection and analysis, decision to publish, or preparation of the manuscript.

==============================
Background

In injury or infection, monocytes migrate into the affected tissues from circulation and differentiate into macrophages which are subsequently involved in the inflammatory responses. Macrophage differentiation and activation have been studied in response to multiple chemokines and cytokines. However, mechanical, and physical stimuli can also influence macrophage differentiation, activation, cytokine production, and phagocytic activity.

Methods

In this study the macrophage differentiation from THP-1 monocytes was assessed upon the stimulation with 1,25-dihydroxyvitamin D3 and 1,000 Hz vibrations, using qPCR for quantification of transcript expression. Vitamin D binds the vitamin D receptor (VDR) and subsequently modulates the expression of a variety of genes in monocytes. The effects of the 1,000 Hz vibrational stimulation, and the combined treatment of vitamin D3 and 1000 Hz vibrations were unknown. The differentiation of macrophages was assessed by looking at transcription of macrophage markers (e.g., CD14, CD36), antigen presenting molecules (e.g., HLA-DRA), transcription factors (e.g., LEF-1, TCF7L2), and mechanosensors (e.g., PIEZO1 and PKD2).

Results

The results showed that vitamin D3 induced THP-1 macrophage differentiation, which was characterized by upregulation of CD14 and CD36, downregulation of HLA-DRA, upregulation of the PKD2 (TRPP2), and an inverse relationship between TCF7L2 and LEF-1, which were upregulated and downregulated respectively. The 1,000 Hz vibrations were sensed from the cells which upregulated PIEZO1 and TCF3, but they did not induce expression of genes that would indicate macrophage differentiation. The mRNA transcription profile in the cells stimulated with the combined treatment was comparable to that of the cells stimulated by the vitamin only. The 1,000 Hz vibrations slightly weakened the effect of the vitamin for the regulation of CD36 and HLA-DMB in the suspension cells, but without causing changes in the regulation patterns. The only exception was the upregulation of TCF3 in the suspension cells, which was influenced by the vibrations. In the adherent cells, the vitamin D3 cancelled the upregulating effect of the 1,000 Hz vibrations and downregulated TCF3. The vitamin also cancelled the upregulation of PIEZO1 gene by the 1,000 Hz vibrations in the combined treatment.

Conclusion

The mechanical stimulation with 1,000 Hz vibrations resulted in upregulation of PIEZO1 in THP-1 cells, but it did not affect the differentiation process which was investigated in this study. Vitamin D3 induced THP-1 macrophage differentiation and could potentially influence M2 polarization as observed by upregulation of CD36 and downregulation of HLA-DRA. In addition, in THP-1 cells undergoing the combined stimulation, the gene expression patterns were influenced by vitamin D3, which also ablated the effect of the mechanical stimulus on PIEZO1 upregulation.

Introduction

Macrophages play important roles in health and disease through phagocytosis of pathogenic microorganisms, by releasing inflammatory mediators, by inducing and maintaining inflammation, and by removing apoptotic cells and repairing tissues (Gordon, 2007; Mosser & Edwards, 2008). Tissue-resident macrophages, derived from the yolk sac at the embryonic stage, are replicated in tissues to maintain cell number, and have different morphology and function depending on the tissue where they reside (Lavin et al., 2015). However, in the case of tissue injury or infection, monocytes derived from bone marrow circulating in peripheral blood, migrate to the affected tissues where they differentiate into macrophages, and are subsequently involved in the inflammatory response (Shi & Pamer, 2011).

THP-1 cells are human immortalized monocytes derived from acute monocytic leukemia and have been extensively used to study macrophage differentiation, functions, signaling pathways, and nutrient and drug transport (Chanput, Mes & Wichers, 2014; Bosshart & Heinzelmann, 2016; Nurminen, Seuter & Carlberg, 2019). In this study we investigated the THP-1 responses towards stimulation with vitamin D3 (1,25-dihydroxyvitamin D3), 1,000 Hz nanovibrations or both, in order to study the expression of genes that could indicate differentiation or mechanosensitivity changes in these cells. In the following text, the combined treatment refers to the combination of 50 nM vitamin D3 and 1,000 Hz vibrations. The cell responses were investigated separately in adherent and suspension THP-1 monocytes upon each stimulation, to avoid averaging results for both the cell types within the same population and consider their differences (Fig. S2).

Vitamin D has been shown to promote monocyte differentiation into macrophages and targets multiple genes (Nurminen, Seuter & Carlberg, 2019). The active form of vitamin D, 1,25-dihydroxyvitamin D3, is a lipophilic molecule which easily passes through biological membranes and binds with high-affinity to the receptor and transcription factor vitamin D receptor (VDR), which is primarily located in the nucleus (Haussler et al., 2013). The activation of vitamin D target genes is explained by the chromatin model (Nurminen, Seuter & Carlberg, 2019). The ligand-activated VDR molecules bind to a wide variety of enhancer regions that carry suitable binding motifs and are located within accessible chromatin. With the help of pioneer factors, such as PU.1, CEBPA, and GABPA, VDR increases the accessibility of chromatin at and around these enhancer regions (Seuter, Neme & Carlberg, 2017; Seuter, Neme & Carlberg, 2018; Nurminen et al., 2019). In THP-1 cells, 1,25-dihydroxyvitamin D3 stimulation significantly affects the binding strength of transcription factor CTCF to topologically associating domain (TAD) anchors, which results in about 600 TADs becoming sensitive to vitamin D (Neme, Seuter & Carlberg, 2016). Looping of activated DNA-bound VDR to a transcription start site (TSS) at these promoter regions results in increased chromatin accessibility as well as of H3K27ac and H3K4me3 marks (Seuter, Neme & Carlberg, 2016; Nurminen et al., 2019). All these vitamin D-triggered changes in the local chromatin structure at enhancer and promoter regions finally lead to the activation of RNA polymerase II assembled on the respective TSSs and the start of mRNA synthesis. The vitamin may also affect gene expression by increasing the expression and the activity of transcription factors other than VDR, such as BCL6, NFE2, POU4F2, and ELF4 (Nurminen et al., 2015).

The effects of vitamin D have been studied in the context of macrophage differentiation from monocytes, but they are also extended into the effector macrophage responses (Hewison, 2010). In fact, normal human macrophages are able to synthesize 1,25-dihydroxyvitamin D3 when stimulated with interferon gamma (IFNγ) (Phillip Koeffler et al., 1985). The vitamin D is involved in the regulation of T cell and macrophage effector functions, primarily via localized autocrine or paracrine synthesis of 1,25-dihydroxyvitamin D3 from its precursor 25-hydroxyvitamin D3 (Hewison, 2010). In addition, vitamin D deficiency is prevalent in multiple autoimmune diseases, such as multiple sclerosis, type 1 diabetes, systemic lupus erythematosus, and alopecia areata, and it is highly associated with the risk of autoimmunity (Yang et al., 2013; Lin, Meng & Song, 2019). Vitamin D has been implicated in prevention and protection from autoimmune diseases by immunomodulation of macrophage, dendritic cell, and T cell responses (Hewison, 2010; Yang et al., 2013).

In the recent years, interest has been given to the mechanobiology of macrophages, which like other immune cells have evolved mechanisms to perceive and respond to the mechanical forces around them (Kim et al., 2019). The cellular functions of tissue-resident macrophages and monocyte-derived macrophages are affected by the tissue-specific microenvironment, which can create many types of mechanical stress on cells (McWhorter, Davis & Liu, 2015; Mennens, Van den Dries & Cambi, 2017). Stiffness and topography, which are mechanical properties of the extracellular matrix, regulate the differentiation, proliferation, and function of macrophages such as phagocytosis (Patel et al., 2012). In monocytes, the PIEZO1 mechanotransduction in response to cyclical hydrostatic pressure, results in HIF1α stabilization and secretion of molecules, such as endothelin-1 (EDN1), and neutrophil chemoattractant CXCL2 (Solis et al., 2019). In addition, macrophages in tissues are exposed to alterations of pressure which affect the secretion of cytokines such as IL-6, TNF-α and IL-1β, (Ferrier, McEvoy & Andrew, 2000; Mevoy et al., 2002). Other mechanical forces that these cells experience originate from dynamic mechanical loading, such as continuous and cyclic stretch and compression (McWhorter, Davis & Liu, 2015; Mennens, Van den Dries & Cambi, 2017).

Just like normal monocytes, THP-1 cells have shown to respond to mechanical stressors. For example, in models of atherosclerosis, biomechanical strain on THP-1 cells can induce expression of the class A scavenger receptor, an important lipoprotein receptor in atherogenesis (Yamamoto, Ikeda & Shimada, 2003). In addition, DNA microarray analysis has shown that cyclic mechanical strain in THP-1 cells induces expression of genes, some encoding for inflammatory markers such as IL-8 and IEX-1(Yamamoto, Ikeda & Shimada, 2003). In these cells, biomechanical deformation influences the degradation of extracellular matrix, monocyte differentiation, and promotion of atherosclerosis (Yamamoto, Ikeda & Shimada, 2003). In addition, as THP-1 cells differentiate they become adherent, a process which may result in altered mechanosensitivity (Tsuchiya et al., 1982; Schwende et al., 1996).

In this study, the cells were mechanically stimulated using 1,000 Hz vibrations. The vibrational stimulation of 1,000 Hz frequency and nano-scale amplitude has been used to study in vitro osteogenic differentiation with successful results (Nikukar et al., 2013; Pemberton et al., 2015; Tsimbouri, 2015; Robertson et al., 2018), and in this study it was used to investigate any effect it may have on the differentiation of macrophages from THP-1 monocytes. Assessment of macrophage differentiation in response to externally applied vibrational stimuli can provide insights into monocyte mechanosensitivity and enquire the therapeutic effects of vibrational treatments in inflammatory diseases.

From a technical point of view, the experiments of this study were designed to give an insight into the differentiation process of THP-1 monocytes into macrophages under different stimulation parameters, compare between treatments, and look into mechanosensor mRNA expression.

Materials & Methods

THP-1 monocyte growth

THP-1 cells (ATCC® TIB-202™) were reconstituted from −80 °C storage and allowed to recover for 2 weeks in cultures, splitting when confluency reached around 8 × 105 cells/mL. The culture medium needed for cell growth was composed of RPMI-1640 with L-glutamine (Capricorn Scientific; RPMI-HA), 10% Foetal Bovine Serum (FBS) (Gibco; A3160802) and 1% Antibiotic-Antimycotic 100X mix (Gibco; 15240062). The cells were cultured at 37 °C, 5% CO2 until ready for the experiments.

Experimental set up

The THP-1 cells were collected from T75 flasks (25 mL suspension) and pelleted by centrifugation at 1,500 rpm for 10 min. The experiment involved 4 replicates of untreated cells, 4 replicates of cells treated with 50 nM 1,25-dihydroxyvitamin D3 (Sigma-Aldrich, D1530), 4 replicates of cells treated with 1,000 Hz vibrations (amplitude range of 30 - 60 nm), and 4 replicates of cells treated simultaneously with 50 nM 1,25-dihydroxyvitamin D3 and 1,000 Hz stimulation. The cells underwent stimulation for 3 days (72 h). No medium or vitamin D3 replacement occurred for this duration of time. The cell density per each replicate at the start of the experiment was 1.5 × 105 cells/mL, in one mL suspension plated on 24-well plates (Thermo Fisher Scientific; 142475). The experiments took place at 37 °C, 5% CO2, and 95% air incubator (LEEC 190D CO2).

Preparation of the vibrational device

Plates (24-well plates) which would be clamped on the bioreactor had magnet sheets (First4Magnets; D-F4MA43MHP) attached 48 h before the start of the experiment, for better adhesion and removal of air pockets with time. In addition, the vibrational device (nicknamed Nanokicking bioreactor) was incubated at 37 °C for 2 days prior to the start of experiments, which was the temperature at which the bioreactor was calibrated. Incubation prior to the experiment was also useful for avoiding condensation upon immediate translocation of the bioreactor from room temperature to incubator environment. The experiments took place in fanless incubator LEEC 190D to avoid additional external vibrations. The bioreactor’s stability and generated vibrations were assessed using laser interferometry every 3 months (diagram in Fig. S1). The platform of the bioreactor was generating vibrations of 1, 000 Hz frequency and amplitude range 30–60 nm at the time of the experiments.

RNA extraction

The RNA was extracted separately for the suspension and adherent cells. Cell suspension was slowly removed and added to sterile RNase-free 1.5 mL tubes. The cells in suspension were pelleted by centrifugation at 3,000 rpm for 5 min. The supernatant was discarded and 1 mL Trizol reagent (Invitrogen; AM9738) was added to homogenize the pellet. For the adherent cells, 1 mL Trizol reagent was added directly in the wells. The lysed cells were homogenized using a 25 g syringe. The RNA extraction from the lysed cells in Trizol solution was done by separating the aqueous phase after addition of 0.2 mL chloroform and centrifugation at 13,000 rpm for 15 min at 4 °C. The RNA was washed with isopropanol and 75% ethanol and stored in 30 µL of nuclease-free water (Gibco; 10977035). Quantification of the RNA in ng/µL was done on Nanodrop 1000, using the RNA nucleic acid program.

DNase treatment

The DNase treatment was performed following the protocol of DNA-free Kit (Thermo Fisher Scientific, AM1906), in order to degrade any genomic DNA that contaminated the RNA solutions during extraction. The maximum RNA concentration for each sample was 5 µg per 50 µL DNase reaction. Removal of genomic DNA contamination allowed efficient detection of amplification during the real-time PCR.

Complementary DNA synthesis

The synthesis of cDNA was done as instructed on the protocol of High-Capacity cDNA Reverse transcription Kit (Applied Biosystems; 4368814). The reaction was comprised of 10 µL of 2X RT Mastermix and 10 µL of purified RNA solution from the previous step. Reaction was started by warming at 25 °C for 10 min, followed by incubation at 37 °C for 2 h for the synthesis of the cDNA, and termination of reaction at 85 °C for 5 min. The newly synthesized cDNA was stored at −20 °C until used for PCR reactions.

Real-time PCR

Real-time PCR was used to quantify gene expression in adherent and suspension THP-1 cells. The PCR amplifications were performed in 25 µL reactions containing 12.5 µL PowerUP SYBR Green Mastermix (Applied Biosystems; A25742); 0.5 µL Forward Primer and 0.5 µL Reverse Primer for the respective genes, 1µL of cDNA and topped up to 25µL with nuclease free water (Gibco; 10977035).

The primer pair used for amplification of the housekeeper RPL37A were RPL37A forward 5′-ATTGAAATCAGCCAGCACGC-3′and RPL37A reverse 5′-AGGAACCACAGTGCCAGATCC-3′. The primer pair used for amplification of the housekeeper ACTB were ACTB forward 5′-ATTGCCGACAGGATGCAGAA-3′and ACTB reverse 5′-GCTGATCCACATCTGCTGGAA-3′. The primer pair used for amplification of CD36 were CD36 forward 5′-TCACTGCGACATGATTAATGGTACA-3′and CD36 reverse 5′-ACGTCGGATTCAAATACAGCATAGAT-3′. The primer pair used for amplification of CD14 were CD14 forward 5′-ACGCCAGAACCTTGTGAGC-3′and CD14 reverse 5′-GCATGGATCTCCACCTCTACTG-3′. The primer pair for amplification of HLA-DRA were HLA-DRA forward 5′-TAAGGCACATGGAGGTGATG-3′and HLA-DRA reverse 5′-GTACGGAGCAATCGAAGAGG-3′. The primer pair used for amplification of HLA-DMB were HLA-DMB forward 5′-CTCTCACAGCACCTCAACCA-3′and HLA-DMB reverse 5′-TAGAAGCCCCACACATAGCA-3′. The primer pair used for amplification of PIEZO1 were PIEZO1 forward 5′-CATCTTGGTGGTCTCCTCTGTCT-3′and PIEZO1 reverse 5′-CTGGCATCCACATCCCTCTCATC-3′. The primer pair used for detection of PKD1 were PKD1 forward 5′-CGCCGCTTCACTAGCTTCGAC-3′and PKD1 reverse 5′-ACGCTCCAGAGGGAGTCCAC-3′. The primer pair used for amplification of PKD2 were PKD2 forward 5′-GCGAGGTCTCTGGGGAAC-3′and PKD2 reverse 5′-TACACATGGAGCTCATCATGC-3′. The primer pair used for amplification of NFAT2 were NFAT2 forward 5′-CACTCCTGCTGCCTTACACA-3′and NFAT2 reverse 5′-AAGATGCGAGCATGCGACTA-3′. The primer pair used for amplification of TCF3 were TCF3 forward 5′-TGACCTCCTGGACTTCAGC-3′and TCF3 reverse 5′-ACCTGAACCTCCGAACTGC-3′. The primer pair used for amplification of TCF4 were TCF4 forward 5′-AGTGCGATGTTTTCACCTCC-3′and TCF4 reverse 5′-CCTGAGCTACTTCTGTCTTC-3′. The primer pair used for the amplification of TCF7L2 were TCF7L2 forward 5′-CCGGGAAAGTTTGGAAGAAG-3′and TCF7L2 reverse 5′-ACTGAAAATGGAGGGTTCGG-3′. The primer pair used for amplification of LEF-1 were LEF-1 forward 5′-GACAGTGACCTAATGCACGT-3′and LEF-1 reverse 5′-CCACCTTCTGCCAAGAATCT-3′.

The primers for TCFs and LEF-1 transcription factors were designed and tested by Dr. Robin Freeburn. Primers amplifying PIEZO1 were designed using the NCBI primer design tool for the mRNA sequence NM_001142864.4, and primers amplifying CD14 were designed similarly for the mRNA sequences NM_001174105.2 (CD14 mRNA transcript variant 4), NM_001040021.3 (CD14 mRNA transcript variant 2), NM_000591.4 (CD14 mRNA transcript variant 1) and NM_001174104.1 (CD14 mRNA transcript variant 3). The primers for NFAT2 were obtained from Dagna, Pritchett & Lusso (2013), primers for HLA-DRA and HLA-DMB were obtained from Ulbricht et al. (2012), primers for PKD1 and PKD2 were obtained from Dalagiorgou et al. (2013), and primers from CD36, ACTB and RPL37A were obtained from Maeß, Sendelbach & Lorkowski (2010).

The efficiency of primers taken from existing literature has been assessed in published papers (Fukuda, Mitsuoka & Schmid-Schönbein, 2004; Maeß, Sendelbach & Lorkowski, 2010; Ulbricht et al., 2012; Dagna, Pritchett & Lusso, 2013; Dalagiorgou et al., 2013). The primer efficiency was assessed prior to the experiments and was around 97% for all the investigated genes. Similar PCR efficiency for each primer is necessary for relative quantification using the ΔΔCT method (Livak & Schmittgen, 2001). The PCR efficiency was also assessed by melt curve analysis. The collected CT values were used for the ΔΔCT relative quantification of expression, comparing the stimulated cells to the untreated controls. The ΔCT was obtained by comparison of CTs of genes of interest to the mean CT of two housekeeping genes RPL37A and ACTB. These housekeeping genes are considered to be the best for the analysis of RNA expression in THP-1 cells (Maeß, Sendelbach & Lorkowski, 2010).

Statistical analysis

The gene expression data are presented as mean of four replicates ± SEM, with little exception where some particular genes were not detected in all replicates. The analysis of statistical significance between the stimulated cells versus controls, and between each type of stimulation was done using unpaired T test with Welch’s correction. Statistical analysis was carried out using GraphPad Prism® version 6. P values < 0.05 were accepted as significant.

Results

Regulation of genes encoding macrophage markers and antigen presenting molecules

Stimulation with vitamin D3, which also served as a positive control for the induction of differentiation, resulted in upregulation of the CD14 and CD36 mRNA in both adherent and suspension cells (Figs. 1A and 2A). Vitamin D3 also downregulated the HLA-DRA expression in adherent and suspension cells (Figs. 1A and 2A). The mRNA of HLA-DMB was upregulated for vitamin D3 stimulation in suspension cells (Fig. 1A). The HLA-DMB was not regulated in response to the vitamin D3 in the adherent cells (Fig. 2A).

Figure 1 Gene expression in response to different stimulations in suspension THP-1 cells, compared to the unstimulated suspension THP-1 cells.

(A) mRNA regulation in response to stimulation with 50nM of 1,25-dihydroxyvitamin D3. (B) mRNA regulation in response to 1,000 Hz vibrations (amplitude 30–60 nm). (C) mRNA regulation in response to the combined vitamin D (50 nM) and 1,000 Hz (30–60 nm amplitude) vibrations. Data presented as mean of four replicates ± SEM. Statistical analysis between stimulated and control values was assessed by unpaired T test with Welch’s correction. P values lower than 0.05 were considered statistically significant.

Figure 2 Gene expression in response to different stimulations in adherent THP-1 cells, compared to the unstimulated adherent THP-1 cells.

(A) mRNA regulation in response to stimulation with 50nM of 1,25-dihydroxyvitamin D3. (B) mRNA regulation in response to 1,000 Hz vibrations (amplitude 30–60 nm). (C) mRNA regulation in response to the combined vitamin D (50 nM) and 1,000 Hz (30–60 nm amplitude) vibrations. Data presented as mean of four replicates ± SEM. Statistical analysis between stimulated and control values was assessed by unpaired T test with Welch’s correction. P values lower than 0.05 were considered statistically significant.

The 1,000 Hz stimulation caused upregulation of CD36 and downregulation of HLA-DMB in suspension cells (Fig. 1B), whereas in adherent cells it only downregulated the HLA-DRA (Fig. 2B).

The combined stimulation induced upregulation of CD14 and CD36 in both adherent and suspension cells (Figs. 1C and 2C). The HLA-DRA was downregulated in both cell types compared to the respective unstimulated control (Figs. 1C and 2C), whereas HLA-DMB was upregulated in suspension cells (Fig. 1C).

The expression values of CD14, CD36, HLA-DRA and HLA-DMB in stimulated cells versus controls are shown in Tables 1 and 2, for suspension and adherent cells respectively.

Table 1 Expression of genes in stimulated suspension THP-1 cells compared to the unstimulated suspension cells at 72 h.

Statistical analysis was performed using unpaired T test with Welch’s correction. Fold change (2−ΔΔCt) values higher than 1 indicate upregulation, whereas values between 0 and 1 indicate downregulation of mRNA transcripts in stimulated cells.

THP-1 cells in suspension	
Stimulation	mRNA	Roles	Fold change (2−ΔΔCT) Stimulated cells vs Control	P value	
50 nM 1,25(OH)2 D3 (72 h)	CD14	Macrophage marker	573.92	0.0161	
CD36	Macrophage marker	3.66	0.0004	
HLA-DRA	Antigen presentation	0.16	0.0001	
HLA-DMB	Antigen presentation	2.17	0.0016	
PKD2	Mechanosensory non-selective cation channel	2.32	0.0004	
TCF4	Transcription factor (unknown roles in macrophages)	0.5	0.001	
TCF7L2	Transcription factor (proliferation and differentiation )	1.85	0.0052	
LEF-1	Transcription factor (proliferation and differentiation )	0.37	0.0005	
Vibrations 1,000 Hz (30–60 nm) (72 h)	CD36	Macrophage marker	1.43	0.013	
HLA-DMB	Antigen presentation	0.86	0.0404	
PIEZO1	Mechanosensory channel	1.39	0.0441	
TCF3	Transcription factor (unknown roles in macrophages)	1.69	0.0182	
50 nM 1,25(OH)2 D3 + Vibrations 1, 000 Hz (30–60 nm) (72 h)	CD14	Macrophage marker	428.9	0.0359	
CD36	Macrophage marker	2.66	0.0009	
HLA-DRA	Antigen presentation	0.19	<0.0001	
HLA-DMB	Antigen presentation	1.51	0.0071	
	PKD2	Mechanosensory non-selective cation channel	2.38	0.0021	
	TCF3	Transcription factor (unknown roles in macrophages)	1.27	0.0142	
	TCF4	Transcription factor (unknown roles in macrophages)	0.55	0.0026	
	NFAT2	Transcription factor (undefined roles in macrophages)	0.69	0.0434	
	TCF7L2	Transcription factor (proliferation and differentiation )	2.1	0.0063	
	LEF-1	Transcription factor (proliferation and differentiation )	0.66	0.0011	

Table 2 Expression of genes in stimulated adherent THP-1 cells compared to the unstimulated adherent cells at 72 h.

Statistical analysis was performed using unpaired T test with Welch’s correction. Fold change (2−ΔΔCt) values higher than 1 indicate upregulation, whereas values between 0 and 1 indicate downregulation of mRNA transcripts in stimulated cells.

THP-1 cells adhered	
Stimulation	mRNA	Roles	Fold change (2−ΔΔCt) Stimulated cells vs Control	P value	
50 nM 1,25(OH)2 D3 (72 h)	CD14	Macrophage marker	650.9	0.0026	
CD36	Macrophage marker	3.13	0.0073	
HLA-DRA	Antigen presentation	0.09	0.0011	
PKD2	Mechanosensory non-selective cation channel	2.47	0.0096	
TCF3	Transcription factor (unknown roles in macrophages)	0.63	0.0134	
TCF4	Transcription factor (unknown roles in macrophages)	0.48	0.0032	
NFAT2	Transcription factor (undefined roles in macrophages)	0.52	0.0458	
TCF7L2	Transcription factor (proliferation and differentiation )	2.63	0.0313	
LEF-1	Transcription factor (proliferation and differentiation )	0.3	0.0491	
Vibrations 1,000 Hz (30–60 nm) (72 h)	HLA-DRA	Antigen presentation	0.07	0.0022	
PIEZO1	Mechanosensory channel	11.44	0.0247	
PKD2	Mechanosensory non-selective cation channel	0.6	0.0236	
NFAT2	Transcription factor (undefined roles in macrophages)	0.12	0.0004	
TCF3	Transcription factor (unknown roles in macrophages)	4.73	0.04	
50 nM 1,25(OH)2 D3 + Vibrations 1,000 Hz (30–60 nm) (72 h)	CD14	Macrophage marker	542.09	0.0011	
CD36	Macrophage marker	2.27	0.0227	
HLA-DRA	Antigen presentation	0.09	0.002	
PKD2	Mechanosensory non-selective cation channel	1.95	0.0013	
TCF3	Transcription factor (unknown roles in macrophages)	0.69	0.0232	
TCF4	Transcription factor (unknown roles in macrophages)	0.37	0.0016	
NFAT2	Transcription factor (undefined roles in macrophages)	0.31	0.0006	
TCF7L2	Transcription factor (proliferation and differentiation)	2.36	0.0116	

A comparison between the treatments was performed for the above genes in suspension (Fig. 3) and adherent cells (Fig. 4). The comparison between treatments is shown in detail in Table 3.

Figure 3 Comparison of fold change values for different genes between the treatments in stimulated THP-1 suspension cells.

Each treatment values were compared to the others using unpaired T test with Welch’s correction. P values lower than 0.05 were considered statistically significant. Genes investigated encode for markers of macrophage differentiation (A), transcription factors (B), and mechanosensors (C).

Figure 4 Comparison of fold change values for different genes between the treatments in stimulated THP-1 adherent cells.

Each treatment values were compared to the others using unpaired T test with Welch’s correction. P values lower than 0.05 were considered statistically significant. Genes investigated encode for markers of macrophage differentiation (A), transcription factors (B), and mechanosensors (C).

Table 3 Comparison of THP-1 gene expression between different treatments.

The arrows indicate upregulation or downregulation of the genes when comparing the different stimuli. Statistical analysis was performed using unpaired T test with Welch’s correction.

mRNA	50 nM Vitamin D3 vs 1,000 Hz	50 nM Vitamin D3 vs 50bnM Vitamin D3+1,000 Hz	50 nM Vitamin D3+1,000 Hz vs 1,000 Hz	
	Adherent	Suspension	Adherent	Suspension	Adherent	Suspension	
CD14	↑ 559.5 Fold
(p = 0.003)	↑ 564.9 Fold
(p = 0.016)	No difference
(p = 0.24)	No difference
(p = 0.41)	↑ 466.0 Fold
(p = 0.001)	↑ 422.2 Fold
(p = 0.036)	
CD36	↑ 9.28 Fold
(p = 0.004)	↑ 2.55 Fold
(p = 0.0003)	No difference
(p = 0.14)	↑ 1.38 Fold
(p = 0.008)	↑ 6.73 Fold
(p = 0.006)	↑ 1.85 Fold
(p = 0.002)	
HLA-DRA	No difference
(p = 0.57)	↓ 0.15 Fold
(p = 0.0001)	No difference
(p = 0.87)	No difference
(p = 0.14)	No difference
(p = 0.39)	↓ 0.18 Fold
(p < 0.0001)	
HLA-DMB	No difference
(p = 0.58)	↑ 2.51 Fold
(p = 0.0008)	No difference
(p = 0.09)	↑ 1.43 Fold
(p = 0.006)	No difference
(p = 0.61)	↑ 1.75 Fold
(p = 0.002)	
NFAT2	No difference
(p = 0.075)	No difference
(p = 0.14)	No difference
(p = 0.28)	No difference
(p = 0.15)	No difference
(p = 0.052)	↓ 0.84 Fold
(p = 0.02)	
TCF3	↓ 0.13 Fold
(p = 0.031)	↓ 0.58 Fold
(p = 0.013)	No difference
(p = 0.44)	↓ 0.77 Fold
(p = 0.045)	↓ 0.15 Fold
(p = 0.009)	No difference
(p = 0.08)	
TCF4	↓ 0.18 Fold
(p = 0.032)	↓ 0.54 Fold
(p < 0.0001)	No difference
(p = 0.28)	No difference
(p = 0.13)	↓ 0.14 Fold
(p = 0.029)	↓ 0.59 Fold
(p < 0.0001)	
TCF7L2	No difference
(p = 0.32)	↑ 1.84 Fold
(p = 0.0034)	No difference
(p = 0.63)	No difference
(p = 0.29)	No difference
(p = 0.46)	↑ 2.09 Fold
(p = 0.0047)	
LEF-1	n/a	↓ 0.36 Fold
(p = 0.0014)	No difference
(p = 0.3)	↓ 0.56 Fold
(p = 0.011)	n/a	↓ 0.65 Fold
(p = 0.0027)	
PKD1	No difference
(p = 0.2)	No difference
(p = 0.06)	No difference
(p = 0.15)	No difference
(p = 0.91)	No difference
(p = 0.54)	No difference
(p = 0.055)	
PKD2	↑ 4.09 Fold
(p = 0.0036)	↑ 2.2 Fold
(p < 0.0001)	No difference
(p = 0.16)	No difference
(p = 0.74)	↑ 3.24 Fold
(p = 0.0001)	↑ 2.3 Fold
(p = 0.0039)	
PIEZO1	↓ 0.11 Fold
(p = 0.026)	↓ 0.59 Fold
(p = 0.015)	No difference
(p = 0.42)	No difference
(p = 0.478)	↓ 0.12 Fold
(p = 0.027)	No difference
(p = 0.052)	

CD14 was upregulated only in response to the vitamin D3, as the mRNA levels were comparable to the cells stimulated by the vitamin only (Figs. 3A and 4A). Similarly, the upregulation of CD36 in the adherent cells was only in response to the vitamin D3 in the combined treatment (Fig. 3A). In suspension cells undergoing the combined treatment, the 1,000 Hz stimulation weakened the upregulation of CD36 by the vitamin D3, which was still higher than the upregulation caused by the 1,000 Hz vibrational stimulation alone (Table 3). In the combined treatment, the 1,000 Hz vibrations also weakened the upregulation of HLA-DMB by the vitamin in the suspension cells (Table 3). Interestingly, the HLA-DRA was downregulated from all treatments at the same level in the adherent cells (Fig. 4A), but only the vitamin downregulated this gene in suspension cells (Fig. 3A; Table 3).

Regulation of genes encoding transcription factor

The stimulation with vitamin D3 downregulated NFAT2 and TCF3 in adherent cells (Fig. 2A). The TCF4 and LEF-1 were downregulated in both adherent and suspension cells stimulated with the vitamin (Figs. 1A and 2A). The TCF7L2 mRNA was upregulated in response to the stimulation with vitamin D3 in suspension (Fig. 1A), and adherent cells (Fig. 2A).

The 1,000 Hz vibrational stimulation upregulated TCF3 in both adherent and suspension cells compared to the respective controls (Figs. 1B and 2B). This type of stimulation also downregulated NFAT2 in the adherent cells (Fig. 2B). The mRNA expression of TCF4, TCF7L2 and LEF-1 were not affected by the vibrational stimulation (Figs. 1B and 2B).

The combined stimulation downregulated NFAT2 in both adherent and suspension cells (Figs. 1C and 2C). The TCF3 mRNA was downregulated in the adherent cells (Fig. 2C), but upregulated in the suspension cells (Fig. 1C). The TCF4 was downregulated in both cell types, and TCF7L2 was upregulated in both cell types (Figs. 1C and 2C). The LEF-1 was downregulated in the suspension cells (Fig. 1C).

The expression values of TCF3, TCF4, TCF7L2 and LEF-1 in stimulated cells versus controls are also shown in Tables 1 and 2, for suspension and adherent cells respectively.

A comparison between the treatments was performed for these genes encoding transcription factors in suspension (Fig. 3B) and adherent cells (Fig. 4B), and shown in Table 3.

The NFAT2 mRNA was downregulated in adherent cells for all the treatments, without difference between each other (Fig. 4B). In the suspension cells, the NFAT2 was downregulated only for the combined stimulation (Fig. 3B).

In adherent cells, the TCF3 mRNA was downregulated in response to vitamin D3 but upregulated for the 1,000 Hz stimulation. In the adherent cells, the vitamin cancelled the upregulating effect of the 1,000 Hz vibration and downregulated TCF3, at comparable levels to the cells stimulated with vitamin D3 only (Fig. 4B; Table 3). However, in suspension cells, the TCF3 upregulation was influenced by the 1,000 Hz vibrations, and the mRNA levels were comparable to the cells stimulated with the 1,000 Hz vibrations alone (Table 3; Fig. 3B).

The TCF4 mRNA was downregulated in response to vitamin D3 stimulation in both suspension and adherent cells. In the combined treatment, TCF4 was influenced by the vitamin only. The 1,000 Hz did not have any influence on the expression of this gene neither alone nor in combination with the vitamin (Figs. 3B and 4B). Similarly, the upregulation of TCF7L2 mRNA was influenced only by the vitamin D3 in both adherent and suspension cells, with the 1,000 Hz stimulation having no effect on the cells when applied alone or in combination with the vitamin (Figs. 3B and 4B).

The mRNA for LEF-1 was downregulated in response to vitamin D3 stimulation. In the adherent cells little RNA was obtained for this gene, and no amplification was detected for the 1,000 Hz stimulation (Table 3). This needs to be investigated in the future to explain whether the lack of amplification was due to very low transcripts levels in total RNA, or because of some inhibitory effect that 1,000 Hz vibrations may have. In the suspension cells stimulated with the combined treatment, the 1,000 Hz weakened the downregulating effect of the vitamin D3, however the vitamin influenced the downregulation (Fig. 3B).

Regulation of genes encoding mechanosensors PIEZO1, PKD1 and PKD2

The stimulation with the vitamin D3 resulted in upregulation of PKD2 (TRPP2) mRNA in both adherent and suspension cells. The stimulation with vitamin D3 alone did not affect the expression of PIEZO1 or PKD1 (TRPP1) (Figs. 1A and 2A).

The 1,000 Hz vibrational stimulation resulted in upregulation of PIEZO1 mRNA in both adherent and suspension cells. In the adherent cells, the stimulation downregulated PKD2 mRNA. The vibrational stimulation did not affect PKD1 expression (Figs. 1B and 2B).

The combined treatment resulted in the upregulation of PKD2 mRNA in both adherent and suspension cells. PIEZO1 and PKD1 were not regulated in cells stimulated with the combined treatment (Figs. 1C and 2C).

The expression patterns of PKD2 and PIEZO1 in stimulated cells versus controls are shown in Tables 1 and 2, for suspension and adherent cells, respectively.

A comparison between treatments was performed for these genes encoding mechanosensors in the suspension (Fig. 3C) and adherent cells (Fig. 4C), and shown in Table 3.

The expression of PKD2 was affected only by the vitamin D3, which also cancelled the downregulation effect of the 1,000 Hz in the adherent cells stimulated with the combined treatment (Table 3).

The PIEZO1 upregulation occurred only in response to the stimulation with 1,000 Hz vibrations, but in the combined treatment the vitamin cancelled the upregulating effect of the vibrational stimulation (Table 3). The expression of PKD1 mRNA was not affected by any of the stimulation methods (Table 3).

A comparison of PIEZO1 expression between adherent and suspension cells stimulated with 1,000 Hz vibrations and the combined treatment was performed (Fig. S3). The 1,000 Hz vibrations upregulated PIEZO1 stronger in adherent cells, than in the suspension cells (Fig. S3A). No difference was observed between the adherent and suspension cells stimulated with the combined treatment (Fig. S3B).

Discussion

TCF/LEF pathway and gene nomenclature

TCF/LEF pathway plays roles in monocyte and macrophage differentiation (Thiele et al., 2001). It must be mentioned that some confusion exists about the nomenclature of the TCFs. The mammalian TCF/LEF family comprises of four nuclear factors designated TCF7, LEF1, TCF7L1, and TCF7L2, which are also known as TCF1, LEF1, TCF3, and TCF4, respectively (Hrckulak et al., 2016). Confusion also exists between the nomenclature of genes and the corresponding products. For example, a gene called TCF3 (NCBI gene ID: 6929), also known as E2A, encodes a product that is different from TCF3 encoded from TCF7L1 (NCBI Gene ID: 83439). Similarly, TCF4 (NCBI gene ID: 6925), encodes for TCF4 which is a different protein from the TCF4 encoded from TCF7L2 (NCBI Gene ID: 6934). In this experiment, the mRNA investigated belongs to genes TCF3 (E2A), TCF4 (E2-2), TCF7L2 and LEF-1, with the last two investigated in the context of WNT canonical pathways in monocyte-derived macrophages (Malsin et al., 2019). The pathways which involve TCF3 and TCF4 gene products can be complex and are not elucidated in context of monocyte to macrophage differentiation.

Vitamin D3 induced macrophage differentiation and downregulated HLA-DRA

Vitamin D3 has shown to target multiple monocyte genes and promote monocyte differentiation into macrophages (Nurminen, Seuter & Carlberg, 2019). Similarly, this study demonstrated that vitamin D3 stimulation induced differentiation of THP-1 monocytes into macrophages, when looking at transcriptional regulation of CD14, CD36 and transcription factors TCF7L2 (encoding TCF4) and LEF-1 (Tables 1 and 2).

The stimulation with vitamin D3 resulted in upregulation of the CD14 and CD36 mRNA in both adherent and suspension cells. This pattern of regulation for these two genes was expected to occur during macrophage differentiation from monocytes (Zhang et al., 1994; Maeß, Sendelbach & Lorkowski, 2010). The CD14 is an important marker of the THP-1 differentiation into macrophages which upregulates strongly upon vitamin D3 stimulation (Schwende et al., 1996; Gocek et al., 2012), as was also observed in this study. Furthermore, CD14 and CD36 are primary target genes for vitamin D3 in THP-1 monocytes (Nurminen, Seuter & Carlberg, 2019). The CD14 and CD36 are proteins involved in macrophage functions. CD14 cooperates with Toll-like receptor 4 (TLR4) to mediate the macrophage immune response to bacterial lipopolysaccharide (LPS) (Zanoni et al., 2011).

CD36 is a scavenger receptor which has been associated with M2 polarization and enhanced phagocytosis (Pennathur et al., 2015; Woo et al., 2016).

It has been reported that M2 activation of bone marrow-derived macrophages with IL-4 has resulted in upregulation of CD36 expression, whereas M1 activation with LPS and interferon-γ has resulted in downregulation of the receptor (Pennathur et al., 2015).

During kidney injury, CD36 is an important phenotypic marker of profibrotic M2 macrophages and a key phagocytic receptor for the clearance of apoptotic cells (Pennathur et al., 2015). Similarly, during the resolution phase of stroke, CD36 macrophages have a reparative role through phagocytosis (Woo et al., 2016).

Vitamin D3 downregulated the expression of HLA-DRA in differentiating THP-1 cells (Tables 1 and 2). HLA-DR has been described as an M1 marker, which is upregulated in THP-1 and monocyte - derived macrophages stimulated with IFNγ/LPS, whereas its expression is very low with IL-4/IL-13 stimulation (Yang et al., 2016). The decreased HLA-DR expression in monocytes has also been associated with anti-inflammatory states or immunosuppression. HLA-DR expression is decreased in all monocyte subsets upon IL-10 exposure in vitro and during septic shock (Monneret et al., 2004; Lee et al., 2017), whereas monocytes that have diminished or no HLA-DR expression, called CD14+HLA-DRlo∕neg monocytes, have emerged as important mediators of tumor-induced immunosuppression (Mengos, Gastineau & Gustafson, 2019).

Downregulation of the HLA-DR protein has been observed in primary monocytes treated with vitamin D3 (Tokuda & Levy, 1996), as well as in dendritic cells (Ferreira et al., 2015). In dendritic cells, the downregulation of HLA-DR has been suggested to be part of tolerance processes induced by vitamin D3 signaling (Ferreira et al., 2015).

The upregulation of CD36 and downregulation of HLA-DRA mRNA by vitamin D3 in day 3-differentiating THP-1 macrophages, could indicate predisposition for M2 polarization.

In addition, vitamin D3 stimulation upregulated the mRNA of HLA-DMB in suspension cells. This molecule is important for antigen loading of the MHC class II by removal of CLIP from HLA-DR (Riberdy et al., 1992; Sloan et al., 1995). In one study, HIV-infected THP-1 monocytes had loss of mRNA for HLA-DR, but the mRNAs for HLA-DM continued to be transcribed, showing that genes may have non-corresponding expression patterns (Shao & Sperber, 2002), similar to what was observed in this study.

This study also identified an inverse relationship between TCF7L2 and LEF-1 mRNA regulation during vitamin D3-induced macrophage differentiation. The TCF7L2 (encoding TCF4) in combination with β-catenin forms a complex that regulates expression of genes in monocytes and it is thus involved in the differentiation process (Thiele et al., 2001; Tickenbrock, 2006; Malsin et al., 2019), whereas LEF-1 facilitates nuclear localization of β-catenin and enhances proliferation in acute myeloid leukemia cells, including THP-1 cells (Morgan et al., 2019). Therefore, the downregulation of LEF-1 and the upregulation of TCF7L2 could indicate decreased proliferation and increased differentiation as THP-1 monocytes become macrophages (Schwende et al., 1996; Thiele et al., 2001; Morgan et al., 2019). The inverse relationship of TCF7L2 and LEF-1 has also been related to shifts in differentiation and proliferation states in other cancer cells (Kriegl et al., 2010; Eichhoff, et al., 2011). This pattern of regulation for these two genes can be signature of THP-1 monocyte to macrophage differentiation.

Another transcription factor downregulated in adherent cells in response to the vitamin D3 was NFAT2. The NFATs are important transcription factors for production of proinflammatory cytokines in T and B cells (Macian, 2005), but their roles are not only limited to the adaptive immune cells. It has been showed that the NFATs are required for Toll-like receptor (TLR)-initiated innate immune responses in bone marrow-derived macrophages (Minematsu et al., 2011). In THP-1 monocytes in vitro, the NFAT2 has shown to inhibit the release of high mobility protein box-1 (HMGB1) (Zhao et al., 2016), a proinflammatory protein with roles in inflammation and autoimmunity (Magna & Pisetsky, 2014). The suppression of NFAT2 expression by siRNA has resulted in increased HMGB1 in the supernatant of cells (Zhao et al., 2016). In T cells, 1,25-dihydroxyvitamin D3 and its receptor complex (VDR-RXR) have shown to inhibit NFAT activity (Wöbke, Sorg & Steinhilber, 2014), but its effect on monocytes and NFAT2 mRNA are not known. In this study, the downregulation of NFAT2 mRNA in the adherent cells, which are considered to be in a more advanced stage of differentiation than the suspension cells (Tsuchiya et al., 1982; Schwende et al., 1996) (Fig. S2), could be related to the production of proinflammatory proteins after the maturation of the monocytes into macrophages.

The vitamin D3 stimulation also downregulated TCF3 (encoding E2A) in adherent cells, and TCF4 (encoding E2-2) in both cell types compared to the respective controls. The roles of the products of these genes are not known in monocyte biology and macrophage differentiation, but as demonstrated in this study they are regulatable upon vitamin D3 stimulation.

The stimulation with vitamin D3 had no effect on the regulation of PIEZO1 or PKD1, but it upregulated PKD2 (TRPP2) mRNA in both suspension and adhesion cells. The roles of polycystin 2 (product of PKD2) are not known in THP-1 monocytes, but the results of this study suggest that the PKD2 mRNA upregulation can be signature of vitamin D3-induced differentiation.

Monocyte responses to 1,000 Hz vibrational stimulation

The THP-1 monocytes are responsive to mechanical stressors. Biomechanical strain on THP-1 cells can induce expression of the class A scavenger receptor, degradation of extracellular matrix, monocyte differentiation, and promotion of atherosclerosis (Yamamoto, Ikeda & Shimada, 2003). In addition, DNA microarray analysis has shown that cyclic mechanical strain in THP-1 cells induces expression of genes, some encoding for inflammatory markers such as IL-8 and IEX-1 (Yamamoto, Ikeda & Shimada, 2003). Furthermore, upon differentiation, THP-1 cells become adherent (Tsuchiya et al., 1982; Schwende et al., 1996), which may result in altered mechanosensitivity. This study used 1,000 Hz vibrations as artificially applied mechanical stimulation, in order to study the mechanosensitivity of THP-1 monocytes and assess if it could affect macrophage differentiation.

The vibrational 1,000 Hz stimulation resulted in upregulation of PIEZO1 transcripts in both suspension (Table 1) and adhesion cells (Table 2). PIEZO1 channels are considered professional mechanosensory proteins, capable of sensing and converting mechanical stimuli (Zhong et al., 2018). Little is known about the mechanosensory roles of these channels in monocytes and macrophages. RNA expression analysis presented in cell atlas shows PIEZO1 expressed in monocytes and macrophages, as well as in THP-1 cells (Human Protein Atlas, Cell Type RNA, Piezo1). PIEZO2 expression has not been detected in blood cells, including monocytes, whereas its expression in THP-1 cells is negligible (Human Protein Atlas, Cell Type RNA, Piezo2).

In monocytes, PIEZO1 has shown to signal in response to cyclical hydrostatic pressure, resulting in HIF1α stabilization and secretion of molecules, such as endothelin-1 (EDN1), and neutrophil chemoattractant CXCL2 (Solis et al., 2019). The PIEZO1 signaling to the cyclical pressure has induced inflammation and infiltration of monocytes, which recruit neutrophils in order to clear pulmonary Pseudomonas aeruginosa infection via EDN1 (Solis et al., 2019). In this study, we demonstrated that THP-1 cells upregulate PIEZO1 mRNA in response to 1,000 Hz vibrational stimulation, when applied in isolation. However, the biological significance of such regulation remains to be elucidated. In addition, the PIEZO1 mRNA upregulation in response to the 1,000 Hz vibrations was stronger in the adherent cells that were in contact with the vibrating surface, compared to the floating suspension cells (Fig. S3A). This could indicate potential involvement of mechanotransduction for the regulation of PIEZO1 expression in 1,000 Hz vibrated THP-1 monocytes.

The 1,000 Hz stimulation also caused HLA-DRA downregulation in adherent cells like vitamin D3, but when combined with the vitamin it did not show any synergetic effect (Table 3). Another gene which was upregulated during the stimulation with 1,000 Hz vibrations, was TCF3. This gene was upregulated in both suspension and adherent cells (Table 1; Table 2), but the role of this gene and its products are not known in monocytes. The 1,000 Hz vibrations downregulated the NFAT2 mRNA at the same levels as the vitamin D3 in adherent cells (Table 2), and just like the vitamin it did not regulate this gene in suspension cells (Table 1).

The vibrational stimulation had no effect on the regulation of other transcription factors such as TCF4, TCF7L2 and LEF-1, which were influenced by the vitamin D3 only (Tables 1–3).

The upregulation of PIEZO1 and TCF3 upon the application of the 1,000 Hz stimulation was interesting, but it was not associated with macrophage differentiation, because there was no transcriptional regulation for genes such as CD14, TCF7L2 and LEF-1 which would indicate transition from monocytes to macrophages. The CD36 was upregulated for the 1,000 Hz stimulation in suspension cells. However, in adherent cells, the CD36 mRNA levels were comparable to the unstimulated adherent controls.

The effects of the combined treatment on gene expression and comparison to vitamin D3 and 1,000 Hz vibrations

The combined treatment induced macrophages differentiation, but the process was influenced mostly by the vitamin D3 (Table 3).

The upregulation of CD14 in suspension and adherent cells undergoing the combined treatment was comparable to cells stimulated with vitamin D3 only (Table 3). The CD36 mRNA was upregulated in the adherent cells at comparable level to the cells stimulated with vitamin D3 only. However, in the suspension cells the 1,000 Hz had slightly weakened the upregulation of CD36 by the vitamin D3. The combination of both stimuli resulted in lower mRNA expression than the stimulation with the vitamin, but higher than the stimulation with the 1,000 Hz, hence it could be said that the 1,000 Hz weakened the upregulating effect of the vitamin (Table 3).

Even though when applied in isolation the 1,000 Hz stimulation caused HLA-DRA downregulation in adherent cells at similar levels to vitamin D3, in the combined stimulation it did not show any synergetic effect. The downregulation of HLA-DRA in suspension cells undergoing the combined treatment was comparable to the cells stimulated with the vitamin D3 only, showing that in the combined treatment this gene was influenced only by the vitamin (Table 3).

In the suspension cells undergoing the combined treatment, the 1,000 Hz weakened the upregulation of HLA-DMB by the vitamin. When applied in isolation the 1,000 Hz vibrations downregulated HLA-DMB, however, in the combined treatment the vitamin overshadowed the effect of the vibrational stimulus and caused upregulation (Table 3).

The combined treatment downregulated NFAT2 at comparable levels to both the vitamin D3 and 1,000 Hz treatments when applied alone in the adherent cells. However, in the suspension cells, the 1,000 Hz and the vitamin D3 may have synergistically caused the downregulation of NFAT2 in suspension cells, because the vitamin and the 1,000 Hz did not regulate this gene when applied in isolation.

The regulation of TCF4 and TCF7L2 in the cells stimulated with the combined treatment was comparable to the cells stimulated with vitamin D3, and the 1,000 Hz stimulation had no effect on these genes in the combined treatment (Table 3), similar to when it was applied in isolation (Tables 1 and 2). The 1,000 Hz vibrations however, weakened the downregulating effect that the vitamin had on the mRNA encoding LEF-1 in the suspension cell. In the adherent cells, the LEF-1 mRNA in stimulated with the combined treatment was comparable to the unstimulated controls, but since the mRNA for this gene was not detected in cell stimulated with vibrations only, comparison could not take place (Table 3).

In the presence of the vitamin D3, the effect of 1,000 Hz stimulation on the regulation of PIEZO1 was cancelled in both adherent and suspension cells. Furthermore, in adherent cells, the vitamin D3 cancelled the upregulating effect of 1,000 Hz on the TCF3 and downregulated the gene (Tables 2 and 3). However, in suspension cells the 1,000 Hz stimulation continued to upregulate TCF3 even in the presence of the vitamin (Table 3). This was the only case in which the effects of 1,000 Hz strongly influenced the expression pattern of a gene in the presence of the vitamin.

Conclusions

This study demonstrated that the stimulation with 50nM vitamin D3 for 3 days drives THP-1 macrophage differentiation, as was determined by upregulation of CD14, CD36 and TCF7L2, and downregulation of LEF-1. The differentiation induced by vitamin D3 was accompanied by downregulation of HLA-DRA and upregulation of PKD2 mRNA. Other genes that were regulated during vitamin D3-induced macrophage differentiation included TCF3 and TCF4 in both suspension and adherent cells, and NFAT2 in adherent cells. The upregulation of the mechanosensitive non-selective cation channel PKD2 mRNA could suggest a role during THP-1 macrophage differentiation, whereas the upregulation of CD36 and downregulation of HLA-DRA mRNA could be indicative of predisposition for M2 polarization.

The vibrational stimulation which was used for the mechanical stimulation of cells did not induce the macrophage differentiation process because there was no transcriptional regulation of CD14 and TCF/LEF transcription factors. However, the 1,000 Hz vibrations influenced upregulation of PIEZO1 and TCF3 in both adherent and suspension cells. Furthermore, in adherent cells, the vibrational stimulation downregulated NFAT2 and HLA-DRA at comparable levels to the vitamin D3 stimulated adherent cells. This indicated that while the 1,000 Hz vibrations did not induce differentiation, they induced regulation of genes in the THP-1 cells. However, the biological importance of such response remains to be elucidated.

In the combined treatment, the 1,000 Hz vibrations interfered with the regulation of some genes by the vitamin D3 but without changing their regulation pattern. The only exception was TCF3 in suspension cells stimulated with the combined treatment, which was upregulated by the 1,000 Hz vibrations against the downregulating influence of the vitamin D3. The biological importance of such interference remains to be elucidated. However, the mRNA regulation patterns of the other genes of interest in the combined treatment were in response to vitamin D3 stimulation.

Furthermore, the influence of the 1,000 Hz stimulus in the presence of the vitamin D3 was cancelled (e.g., for PIEZO1 in both cell types), overshadowed (e.g., for CD36 in suspension cells), or cancelled and reversed (e.g., PKD2 in adherent cells). This can have implication for the medicinal application of the 1,000 Hz (nano-scale amplitude) vibrations, because in inflamed tissues rich in chemical signals such as cytokines and chemokines, the cells may lose the ability to sense and respond to such mechanical stimulus.

Further work is necessary to assess the reproducibility of the observations of this study, especially in response to the 1,000 Hz vibrational stimulation. This study was limited by the technology, which was not provided for repeated runs and further work. Increased replicates, expanded time-points, assessment of protein expression, and use of primary monocytes to compare to THP-1 cell responses, are recommended for future work from the authors of this report. In addition, the effects of different frequencies and of vibrations applied in cyclical short-term patterns remain to be studied, in order to expand our understanding of THP-1 cell responses towards the vibrational stimulation.

Overall, this study presents experimental results indicating that the vibrational mechanical forces can be sensed by THP-1 monocytes, but that the chemical ligands such as vitamin D3 remain superior for the induction of macrophage differentiation.

Supplemental Information

Supplemental Information 1 Nanokicking bioreactor (A) and diagram of laser interferometry (B)

(A) Signal generator (left) and the Nanokick bioreactor in the incubator (right). Magnet-clamped plates can be seen on top of the bioreactor’s platform. (B) Laser interferometry was used to measure the frequency and the amplitude of the vibrations on the surface of the wells, which is the site where the mechanical stimulation was applied on the cells. Frequency and the amplitude of the vibrations were measured continuously over periods of 2–3 months, before and after experiments, to allow continuous assessment of the bioreactor’s functionality. The measurements were performed on the bioreactor twice; once after being left at room temperature (25 ° C) and another time after being incubated at 37 ° C for 24 h. During the laser interferometry, a continuous helium-neon beam (wavelength 632.8 nm) is reflected from the surface of the well at a distance of 25–30 cm and directed into the interferometer (SIOS Meßtechnik GmbH SP S-120) to create an interference pattern with a reference beam. Alignment of the beam is achieved by utilising an oscillator signal that appears as a circle on the oscilloscope’s screen. (C) The interferometers output is then analysed by INFAS Vibro computer software (SIOS Meßtechnik GmbH: Interferometry Analysis Software for Vibrometers), which performs a fast fourier transform (FFT) and plot the amplitude of motion in frequency space. The red arrow shows the frequency reading (1 kHz), and the blue arrow shows the amplitude of vibration (32.3 nm).

Click here for additional data file.

Supplemental Information 2 CD14 and CD36 mRNA expression in adherent versus suspension unstimulated THP-1 cells at 72 h culture in complete growth medium

(A) Expression of CD14 mRNA in unstimulated adherent THP-1 cells ( N = 3) was 17.9 fold higher compared to the unstimulated suspension THP-1 cells ( N = 4) ( p value = 0.014). B) Expression of CD36 mRNA in unstimulated adherent THP-1 cells ( N = 3) was 4 fold higher compared to the unstimulated suspension THP-1 cells ( N = 4) ( p value = 0.006).

Statistical analysis was performed using unpaired T test. Fold change was calculated using the ΔΔ Ct method. Fold change values higher than 1 indicate upregulation, whereas values between 0 and 1 indicate downregulation of mRNA transcripts.

These results showed that the unstimulated THP-1 cells that had become adherent at 72 h, expressed higher macrophage markers than the unstimulated cells in suspension. This observation indicated differences between the suspension and adherent cells within the same population. Therefore, in this study the expression of genes upon stimulation was assessed in suspension and adherent cells separately, by comparing to the unstimulated suspension and adherent controls, respectively.

Click here for additional data file.

Supplemental Information 3 Comparison of PIEZO1 mRNA regulation between suspension and adherent THP-1 cells stimulated with 1,000 Hz vibrations and the combined 1,000 Hz and vitamin D3 treatment

(A) PIEZO1 mRNA regulation in response to the 1,000 Hz vibrations. The 1,000 Hz vibrational stimulation upregulated PIEZO1 mRNA in stimulated suspension cells compared to the unstimulated suspension controls ( p∗ = 0.044), and in stimulated adherent cells compared to the unstimulated adherent controls ( p∗ = 0.025). The mean fold change value of PIEZO1 in the stimulated adherent cells was 8.2 fold higher than in the stimulated suspension cells ( p∗ = 0.027). This observation showed that the adherent cells, which were in contact with the vibrating surface of the well, responded by stronger upregulation of PIEZO1 mRNA compared to the floating cells. This could indicate potential involvement of mechanotransduction for the regulation of PIEZO1 expression in vibrated THP-1 monocytes. (B) PIEZO1 mRNA regulation in response to the combined stimulation with 50 nM of 1,25-dihydroxyvitamin D3 and 1,000 Hz vibrations. PIEZO1 mRNA in the stimulated suspension and adherent cells, were comparable to the unstimulated respective controls. No difference was recorded when comparing the fold change values between the stimulated suspension and stimulated adherent cells ( p = 0.061). Even though the application of the 1,000 Hz vibrations in isolation, resulted in strong upregulation of PIEZO1 in adherent cells (A), the vitamin D3 cancelled such effect in the combined treatment (B).

The data presented as mean of four replicates ±SEM, with exception of adherent controls ( N = 3). The statistical analysis was performed using unpaired T test with Welch’s correction. P values lower than 0.05 were considered statistically significant.

Click here for additional data file.

Supplemental Information 4 Raw fold change values of genes in stimulated THP-1 cells versus unstimulated controls

This .pzfx file contain the fold change values of all genes in the stimulated and unstimulated cells. The graphs obtained from these data are presented on Figs. 1 and 2. Statistical analysis in this study was performed between the stimulated cells and unstimulated controls for each gene, using unpaired T test with Welch’s correction. The .pzfx files can be opened in GraphPad Prism.

Click here for additional data file.

Supplemental Information 5 Raw fold change data used for comparison of gene expression in the stimulated cells between the treatments

This .pzfx file contain the fold change values of all genes in the stimulated cells for comparisons between the treatments. The graphs obtained from these data are presented on Figs. 3 and 4. The results of treatment comparisons are presented in Table 3. Statistical analysis in this study was performed using unpaired T test with Welch’s correction, comparing the gene expression values between two different treatments. For example: comparing CD14 fold change values in VitD3 vs 1,000 Hz, VitD3 vs the combined treatments, and 1,000 Hz vs the combined treatment. The .pzfx files can be opened in GraphPad Prism.

Click here for additional data file.

Supplemental Information 6 T test results comparing CD14 and CD36 in unstimulated suspension and adherent cells

The graphs obtained from these data are presented in Fig. 2. Fold change values (2 −ΔΔCt) were obtained using the ΔΔ CT method. The Δ Ct values were obtained by calculating the difference between the Ct of gene of interest to the mean of housekeeping genes ACTB and RPL37A. The ΔΔ Ct values were obtained by calculating the difference between the Δ Ct of the adherent cells vs Δ Ct of the suspension cells. Statistical analysis between the values was performed using unpaired T test. The .pzfx files can be opened in GraphPad Prism.

Click here for additional data file.

Supplemental Information 7 Comparison of PIEZO1 expression between suspension and adherent cells in mechanically stimulated cells

This .pzfx file contain the fold change values of PIEZO1 for comparison between suspension and adherent cells, stimulated with 1,000 Hz vibrations and the combined Vitamin D3 and 1,000 Hz stimulation. The graphs obtained from these data are presented in Fig. 3. Statistical analysis in this study was performed using unpaired T test with Welch’s correction. The .pzfx files can be opened in GraphPad Prism.

Click here for additional data file.

The authors would also like to thank Dr Paul Campsie and Prof Stuart Reid for providing a “Nanokick” bioreactor for the vibrational stimulation of the cells, and for access to the interferometer for the assessment of bioreactor’s stability.

Additional Information and Declarations

Competing Interests

Author Contributions

Data Availability

The authors declare there are no competing interests.

Theodoros Simakou conceived and designed the experiments, performed the experiments, analyzed the data, prepared figures and/or tables, authored or reviewed drafts of the paper, and approved the final draft.

Robin Freeburn and Fiona Henriquez conceived and designed the experiments, analyzed the data, prepared figures and/or tables, authored or reviewed drafts of the paper, and approved the final draft.

The following information was supplied regarding data availability:

The raw fold change values are available in the Supplemental Files.

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
