# Peer review of "Gene expression during THP-1 differentiation is influenced by vitamin D3 and not vibrational mechanostimulation"

_PeerJ, doi:10.7717/peerj.11773_

## Round 0.1 · original submission · Major Revisions

Please provide detailed answers/corrections to the reviewers' comments.

The abstract should be shortened and conclusions should be added.

The discussion needs to be strengthened and provide outlooks.

Please revise the manuscript text for formatting mistakes ("Reference source problem" in the results section). Figure legends should be written more accurately and include information about A), B), etc. figures.

Additional Figure, showing the experimental setup for vibration treatment (including device, plate with cells, and incubator) will simplify the reading/reproducibility of the manuscript and should be added.

·

Basic reporting

This is an interesting but very complicated manuscript. It contained a great deal of information derived from an extensive set of experiments using immortalized cells and the authors tried very hard to organize it in a way that is readable, digestible, and relevant. There are 8 experimental variables (adherent + suspension) x (vibration + vitamin D + vibration and vitamin D + control) with 12 outcome measures belonging to three major categories: monocyte-macrophage differentiation, antigen presentation, and mechano-sensors. Together, there are 96 sets of results. Despite their efforts, this paper reports, and I quote Theodosius Dobzhansky, "a pile of sundry facts -- some of them interesting or curious but making no meaningful picture as a whole".

Experimental design

I do not like reductive experimental designs. Taken out of the local environment, with no supporting extracellular matrix and other cells, these in vitro, ex vivo studies cannot duplicate the complex adaptive system that is present in the tissues or organs. This study uses spontaneously immortalized cell line, THP-1, from a pediatric patient with monocytic leukemia. While this cell line has been extensively used in many laboratories, it is not the same as peripheral blood monocytes and certainly different from the precursors of yolk sac-derived tissue resident macrophages.

Mechanical perturbations are important as cells exist in a mechanical environment and are subjected to tensile, compressive, and shear forces, including, and especially, the tissue resident macrophages. The selection of cyclic loading is good as majority of our daily activities generate stresses that are not static or even quasi-static. However, the selection of 30 to 60 nm amplitude and 1000 Hz frequency demands biological relevance. What kind of daily activities can generate that kind of continuous "nano-kicks" for 3 days?

Regarding the calibration using laser interferometer, was this done within the incubator? How was the vibration administered? That is was it vertical (up-and-down) or horizontal (side-to-side)? Since the authors wish to find out the effect of mechanical forces on the THP-1 cells, what was the force profile with this vibration? How much cellular deformation can occur? a 5 micron (10E-5 m) monocyte with 50 nm (10E-8) deformation give a nominal maximum cellular strain in the 0.001 range, likely much less. What evidence can the authors provide to support that such strain range is physiologically relevant? Compounding the problem is the liquid medium. How much damping can take place? How does the viscosity of the fluid affect the force transmission?

Lastly, it would be interesting to check what cells become adherent out of these immortalized cells which are more like peripheral blood monocytes? How many remain in suspension and how many attached to the plate?

Validity of the findings

I like how the authors presented this set of experimental data, and I believe they are valid. It is the higher level synthesis that is missing. I encourage the authors to focus on the fact that vitamin D3 is the much stronger stimulus than this very artificial vibration protocol. These are largely non-adherent peripheral blood monocytes and as such, they probably normally experience different kind of mechanical loading such as fluid shear, rather than tensile stress or bi-axial compression.

Additional comments

I learned a lot from reviewing your paper, for that I am grateful. One of my mentors from decades ago would ask me, after I proposed my experiments, "Let's fast forward a year or two and say that you have done these experiments and found this and that. So what? Then what?"

I would very much appreciate if the authors would answer the so what and then what.

·

Basic reporting

The figure legend for Figure 4 should be revised. I believe this refers to adherent cell data yet suspension is written.
Figure legend for Table 2 should be revised. I believe this is comparison of stimulated adherent THP-1 cells vs unstimulated adherent cells

Improve English language and sentence structure throughout manuscript e.g., repetitive use of the phrase “the vitamin D3” can simply be replaced with “vitamin D3” for example line 288 “The vitamin D3 downregulated..” instead “Vitamin D3 downregulated..” may be used.

Improve sentence structure in lines 609-610 “This study had limitation because of the technology” – e.g., This study was limited by the technology which…
line 73-74; improve sentence structure. Maybe break the sentence in two

Experimental design

The methodology should be clearer. For example, over the 72-hour period, how was vitamin D3 administered? Is it given once in the medium and left for 3 days? Is it replenished after each day?

Did the authors look at PIEZO2 in THP-1 cells? It may be informative to include the sibling of PIEZO1 for comparative purposes. RT-qPCR analysis of this mechanosensor may be included across vitamin D3 treated, 1000Hz and combined conditions.
Alternatively, the reason why it was not analysed should be presented e.g., future studies will do so; or based on previous data PIEZO2 is not highly expressed in THP1-cells (Reference).

Using only one vibrational parameter (1000Hz) may limit potential mechanosensory responses. For example, previous studies (Koizumi et al, 2014; Functional role for Piezo1 in stretch-evoked Ca2+influx and ATP release in urothelial cell cultures. J Biol Chem 289 (23):16565-16575) have shown the presence of a stretch stimulated threshold that must be attained before mechanosensor mediated changes in cell responses (ATP efflux) are observed. Similarly, the current study may yet to uncover a potential vibrational range that alters gene expression differently than reported. Is there a spectrum of gene expression changes that occurs by changing the range of vibration (Hz, time, cyclical stimulation?). Is there a range in which macrophage differentiation is promoted? Furthermore, the nature of mechanical stimuli (vibration vs stretch vs hydrostatic pressure changes) may alter THP-1 cell`s gene expression. The 1000Hz is a standard range used in THP-1 cells thus it is understandable why the current report did not alter it. However, commenting on these issues may help the reader understand the dynamic range of cell responses that have yet to be elucidated. For the current scope of this report additional experiments are not essential (specially as the authors explained the limitation; lines 609-610), however acknowledgement or dismissal of such potential changes using previously published data or inference from data presented by the study should be presented to the reader.

Validity of the findings

Conclusion should be added to abstract. Currently the abstract abruptly ends on Results. A clear “conclusion:” with a sentence or two to summarise the main findings should be included.

The authors summarise the data elegantly using tables (Tables 1, 2 and 3); however, the main text makes little to no reference of this. Guiding readers to use these tables as comparisons are made may help with the flow of data presentation and discussion points.

Additional comments

In this study Simakou et al, set out to examine the effects of vitamin D3 administration, vibrational stimulation and their combined effect on THP-1 cells with a focus on their differentiation potential into macrophages. A series of RT-qPCRs were conducted to measure the various gene expression changes associated in response to stimuli (vitamin D3 vs 1000Hz or both) and cell condition (suspended vs adherent). The data presented shows novelty and proficiency that should make this manuscript eligible for publication in PeerJ.

The data is efficiently summarised and presented well in both graph and table format which allows clear comparison and interpretation of the data. The authors show novel expression patterns such as PKD2 upregulation with vitamin D3 administration. With further testing in other cell types, this may serve as unique signature of vitamin D3 stimulation.
A comment on potential limitations of THP-1 immortalised cell line vs primary cell lines may be warranted in order to acknowledge the need for further research and confirm findings in the future. In order to enhance readability, the authors should significantly improve sentence structure and avoid grammatical errors. This will avoid breaking the flow of the text and hasten understanding of key points presented.

It would be informative to compare basal expression levels of key genes in suspended and adherent conditions. For example, do adherent cells express comparably higher levels of PIEZO1 than unstimulated suspended cells? Are the levels of CD14 and CD36 higher in adherent vs suspended cells?

Perhaps beyond the scope of the current study, but analysis of protein expression levels of key macrophage differentiation markers (CD14, CD36 etc.) through western blot may strengthen the claims made. The authors could state the need to confirm their key findings at the protein level to corroborate their RT-qPCR data.

Reviewer 3 ·

Basic reporting

The manuscript is well written with sufficient literature support.
The study is an observational study designed to investigate effects of VitD and/or vibration treatment on monocyte to macrophage differentiation. The rationale of the investigation was however unclear; or could have improved by proposing some potential clinical application or translational value.
The discussion reads a bit long on the background support compared to the novel findings. It would be easier to the reader if the focus of each paragraph is shifted more towards the novel findings.

Experimental design

The experimental design is straight forwards with sufficient description for replication. The results are presented with good clarity.

Validity of the findings

The results are well presented and support the conclusions being made.

Additional comments

There are a large number of erroneous table/figure references in the results. Please revisit to fix them.

---

## Round 0.2 · accepted · Accept

Revision has addressed previous questions and there are no further comments.

·

Basic reporting

The manuscript shows drastic improvements in legibility and flow of data presentation. References to tables have been added which improve the reader's ability to follow the claims made. Structurally the manuscript is well organised and self-contained in its discussion in light of the results presented.

Experimental design

Research questions are well designed and addressed efficiently in the experiments chosen and data analysis employed. Many of my original concerns have been addressed and supplementary figures added that further support this manuscript.
Methods are explained and previous issues have been addressed.

Validity of the findings

As stated previously, data presented in this manuscript shows novelty and proficiency. The role of Vitamin D3, vibrational stimulation and THP-1 differential potential is addressed well in this report. The findings should have cross-field relevancy in other cell types.
Conclusion is now referenced in abstract and the authors discuss their findings well. By including supplementary figure 2 the authors elegantly address previous concerns. Overall, the findings presented here and corrections applied should warrant the current manuscript novel and impactful for a wider readership.

Additional comments

An annotated pdf is not necessary. Thank you for addressing the concerns raised. The manuscript presents data to the field of mechanosensation that will intrigue others.

Reviewer 3 ·

Basic reporting

Revision has addressed previous question and I have no further comment.

Experimental design

I have no further comment.

Validity of the findings

I have no further comment.

Additional comments

I have no further comment to the authors for improvement.